# Interventional Cardiovascular Magnetic Resonance Imaging (iCMR) in an Adolescent with Pulmonary Hypertension

**DOI:** 10.3390/medicina56120636

**Published:** 2020-11-24

**Authors:** Surendranath R. Veeram Reddy, Yousef Arar, Tarique Hussain, Gerald Greil, Luis Zabala, Bibhuti B. Das

**Affiliations:** 1Division of Cardiology, Department of Pediatrics, Children’s Medical Center Dallas, UTSW Medical Center, Dallas, TX 75235, USA; Surendranath.Reddy@utsouthwestern.edu (S.R.V.R.); Yousef.Arar@utsouthwestern.edu (Y.A.); Tarique.Hussain@utsouthwestern.edu (T.H.); Gerald.Greil@utsouthwestern.edu (G.G.); Luis.Zabala@utsouthwestern.edu (L.Z.); 2Division of Cardiology, Department of Pediatrics, Baylor College of Medicine, Texas Children’s Hospital, Austin Specialty Care, Austin, TX 78759, USA

**Keywords:** interventional cardiac MRI, pediatric pulmonary hypertension

## Abstract

The interventional cardiac magnetic resonance imaging (iCMR) catheterization procedure is feasible and safe for children and adults with pulmonary hypertension and congenital heart defects (CHD). With iCMR, the calculation of pulmonary vascular resistance (PVR) in children with complex CHD with multilevel shunt lesions is accurate. In this paper, we describe the role of the MRI-guided right-sided cardiac catheterization procedure to accurately estimate PVR in the setting of multiple shunt lesions (ventricular septal defect and patent ductus arteriosus) and to address the clinical question of operability in an adolescent with trisomy 21 and severe pulmonary hypertension.

## 1. Introduction

Pediatric pulmonary hypertension (PH) in children is a complex heterogeneous disease and differs from adults in many respects in etiology and outcomes. Pulmonary arterial hypertension is defined as mean pulmonary artery (PA) pressure >25 mm Hg at rest, with a normal pulmonary artery wedge pressure <15 mm Hg, and an increased pulmonary vascular resistance (PVR) >3 Wood Units × m^2^ [1]. It is a progressive disease that increases the right ventricle’s (RV) workload, causing morphologic change and eventual dysfunction. The RV failure determines the outcomes, and therefore, reliable evaluation of RV function is crucial [2]. Echocardiography is the most commonly used noninvasive tool for screening for PH and serial follow-up [3]. However, accurate estimation of mean PA pressure and PVR is essential for management, not accurately achieved with echocardiography. Right heart catheterization (RHC) remains the gold standard for diagnosing and estimating the severity of PH in children. However, due to cumulative radiation exposure and, most importantly, the unpredictable nature of Fick’s principle-based pulmonary blood flow calculation in complex cardiac conditions with multiple left-to-right shunts, it makes RHC not an ideal tool for PH evaluation and serial follow-up.

Invasive cardiac magnetic resonance (CMR) is useful for comprehensive hemodynamic assessment of congenital heart disease (CHD) patients with multiple shunts [4,5,6]. The first study by Moledina et al. demonstrated that iCMR is feasible in children with PH [7]. Invasive CMR offers a radiation-free examination with comprehensive 3D anatomy, volumetric analysis, and more accurate pulmonary and systemic blood flow calculations [8]. Standard RHC to measure accurate pulmonary blood flow poses a significant challenge in patients with multiple left-to-right shunts as the position of catheter and entrance point of patent ductus arteriosus (PDA) will significantly influence the pulmonary arterial saturation that is plugged into the Fick’s equation for pulmonary blood flow calculation. Any minor change in pulmonary arterial saturations related to the patient’s anatomy or operator technique will significantly influence the total pulmonary blood flow and resultant PVR calculation. The iCMR provides accurate pulmonary blood flow calculations and is independent of confounding variables related to catheter-based measurement. Using the accurate pulmonary blood flow measurements from MRI flow analysis and catheter-measured transpulmonary gradient, an accurate PVR can be calculated, thereby allowing for correct critical decisions regarding the operability of complex congenital heart disease patients [6]. In this paper, we describe the feasibility and usefulness of CMR-guided RHC in an adolescent with trisomy 21 and severe PH in the setting of multiple shunt lesions (ventricular septal defect (VSD), and PDA). This study was approved by the Institutional Review Board (IRB) (STU 032017–061; approved 1 March 2017).

## 2. Case Report

A 16-year-old adolescent boy, trisomy 21 with multiple cardiac shunts (VSD, and PDA) and echocardiography-estimated RV systolic pressure 73 mm Hg plus right atrial pressure based on tricuspid regurgitation jet (Figure 1). He was considered inoperable due to bidirectional shunting across VSD (Figure 2) by echocardiography and referred to our institution for evaluation. He underwent CMR-guided RHC and pulmonary vasodilator testing in the Phillips Ingenia 1.5 Tesla MR/catheter hybrid laboratory suit (Figure 3). The hybrid Cath-CMR setting and iCMR procedural details at Children’s Medical Center, University of Texas Southwestern Medical Center, was described previously [4].

Total pulmonary blood flow assessment was done by summing the flow in the right pulmonary artery (RPA) and the left pulmonary artery (LPA) distal to PDA entrance, the approach based on the assumption that total pulmonary flow can be more accurately quantified by measuring flow in each branch artery [9]. Cardiac output was calculated by measuring blood flow in the ascending aorta as previously described [10]. The total pulmonary blood flow was calculated using the measured RPA and LPA net antegrade blood flow volume [11]. A 6 French balloon wedge catheter was used to perform the CMR-guided RHC procedure. Using the Philips interactive scanning mode guidance (Philips Healthcare), the interventionalist advanced the gadolinium-filled balloon tip of the wedge catheter and MR-conditional guidewire (Emeryglide MRWire Nano4Imaging, Aachen, Germany) (Figure 4). Magnetic resonance imaging provided excellent soft-tissue contrast for completing the right heart catheterization, including 3D reconstruction of the aortic arch, PDA, and branch pulmonary arteries (Figure 5).

The hemodynamics and MRI results obtained in our patient are summarized in Table 1 and Table 2. The pulmonary and systemic cardiac output was also measured by the traditional Fick’s principle method using presumed oxygen consumption, and catheter-measured oxygen saturations obtained at RHC. Based on the hemodynamic and pulmonary blood flow data obtained by traditional Fick’s principle-based calculations, indexed PVR (PVRi) was calculated to be 10.4 Wood unit × m^2^ at baseline and with vasodilator challenge (100% oxygen and 40 ppm nitric oxide) decreased to 9.4 Wood unit × m^2^ in the setting of multiple intracardiac shunts. However, the iCMR PVRi was calculated by dividing the catheter-measured transpulmonary pressure gradient by indexed pulmonary blood flow, calculated accurately by MRI blood flow data, taking into account the various pulmonary blood supply sources related to multilevel left-to-right shunting [6]. The iCMR PVRi was calculated to be 8.7 Wood unit × m^2^ at baseline and decreased to 6.8 Wood unit × m^2^ with vasodilator challenge. The RV size and systolic function were normal (EF 65%), and there was no evidence of scarring. The perimembranous VSD measured 22 × 17 mm with inlet extension to inlet muscular septum and left-to-right shunt. The PDA was large and measured 9 × 8 mm at its narrowest pulmonary end with a left-to-right shunt. Both the VSD and PDA shunt flow increased with vasodilator testing (Table 2). The trachea and main stem bronchi were normal while on positive pressure ventilation. There was mild posterior tracheal flattening. Lung parenchyma was normal. Because the PVRi was 6.8 Wood unit × m^2^ with vasodilator challenge by iCMR calculations, it was decided to proceed with surgery to close the VSD and PDA in our patient.

## 3. Discussion

The MRI-guided RHC in PH is only a small part of iCMR, which has broader applications in interventional procedures, electrophysiological study, and ablation procedures [12,13]. The main advantage of iCMR is that it provides the best of both imaging modalities with direct measurement of the transpulmonary gradient by RHC, and CMR-measured accurate and reliable pulmonary and systemic blood flows [5,6]. With better fluoroscopy equipment and ongoing radiation reduction strategies, the life-time cumulative radiation dose for children and adults has been steadily decreasing. However, no radiation is better than some radiation, and thus MRI-guided RHC may reduce the risk of oncologic risk of medical radiation [14,15]. The other advantage is that we can reliably evaluate airway abnormalities and structural lung disease simultaneously, especially in trisomy 21, as in this report.

Overall, our experience with this case and many CHD patients [4] demonstrates that iCMR can be safely performed in patients with adverse cardiopulmonary hemodynamics and incorporated into routine clinical practice without radiation exposure. Recently, Knight et al. have shown that the procedural time for CMR-RHC is comparable to a standard CMR examination but with the advantage of obtaining the additional hemodynamic information [8]. As with any new imaging or catheterization procedures, proceduralists have a learning curve to get comfortable with iCMR technology, thereby potentially increasing the procedural and anesthetic time, complexity, and associated risks [16]. However, as shown by Knight et al., this is typically an initial/short-term problem only, and the procedural time is significantly shorter after the learning curve, especially as the iCMR teams optimize their departmental clinical protocols.

## 4. Conclusions

Real-time CMR tools are now readily available on all CMR scanners. However, iCMR has been established in the clinical routine in only a few centers worldwide for various indications in children and adults. Broader application of iCMR, including routine evaluation of PH in children, is expected shortly owing to multiple advantages compared to conventional catheterization.

## Figures and Tables

**Figure 1 medicina-56-00636-f001:**
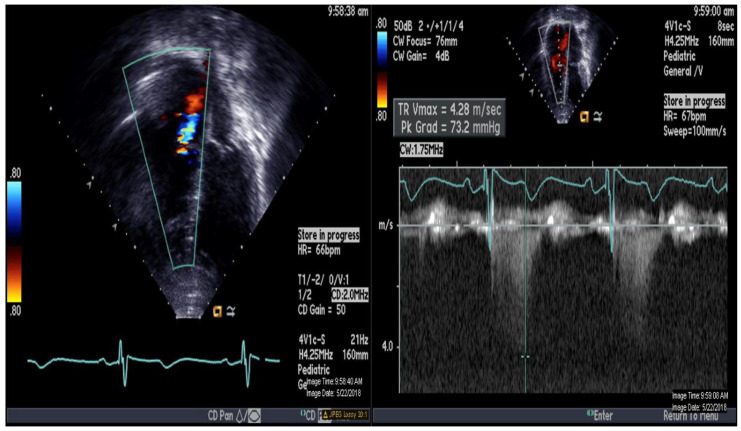
Echocardiography demonstrating apical 4-chamber view with color and continuous Doppler of tricuspid regurgitation jet.

**Figure 2 medicina-56-00636-f002:**
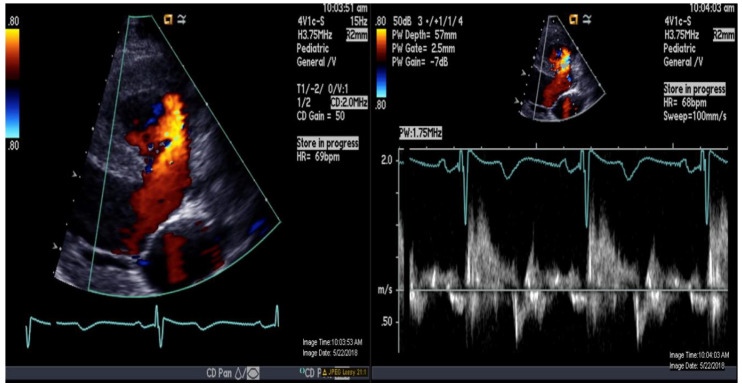
Color-and-pulse Doppler showing bidirectional shunting across a large perimembranous ventricular septal defect.

**Figure 3 medicina-56-00636-f003:**
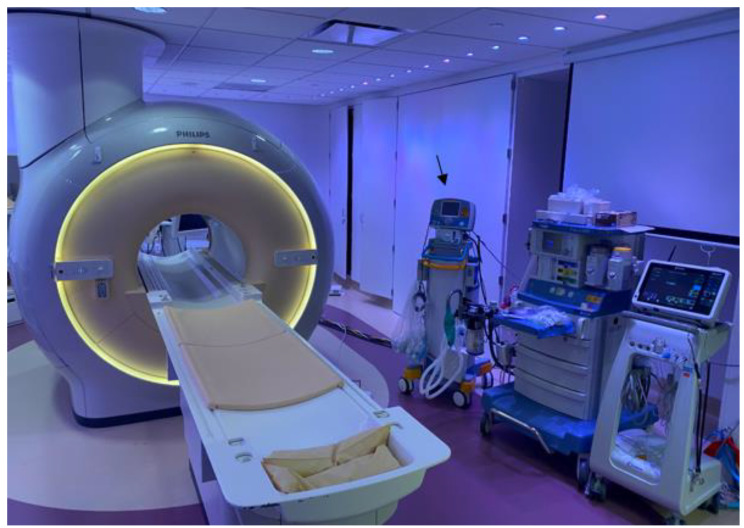
Invasive cardiac magnetic resonance (iCMR) pulmonary hypertension evaluation environment. MR-conditional nitric oxide (INOmax DS_IR_^@^ Plus Delivery Systems, Mallinckrodt Pharmaceuticals, USA) (**arrow**) next to the Philips Ingenia 1.5 Tesla magnet.

**Figure 4 medicina-56-00636-f004:**
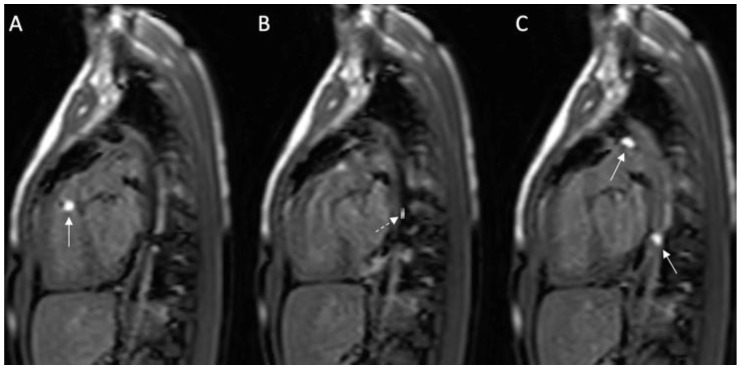
(**A**–**C**): Series of images during the invasive cardiac magnetic resonance (CMR) in this case. (**A**) Gadolinium-filled balloon in right ventricular outflow tract (arrow); (**B**) MR-compatible wire placed prograde across the patent ductus arteriosus (PDA) into the descending aorta; (**C**) Two gadolinium-filled balloons positioned within the PDA and descending aorta. Solid white arrow: gadolinium-filled balloon; dashed white arrow: MR-compatible wire.

**Figure 5 medicina-56-00636-f005:**
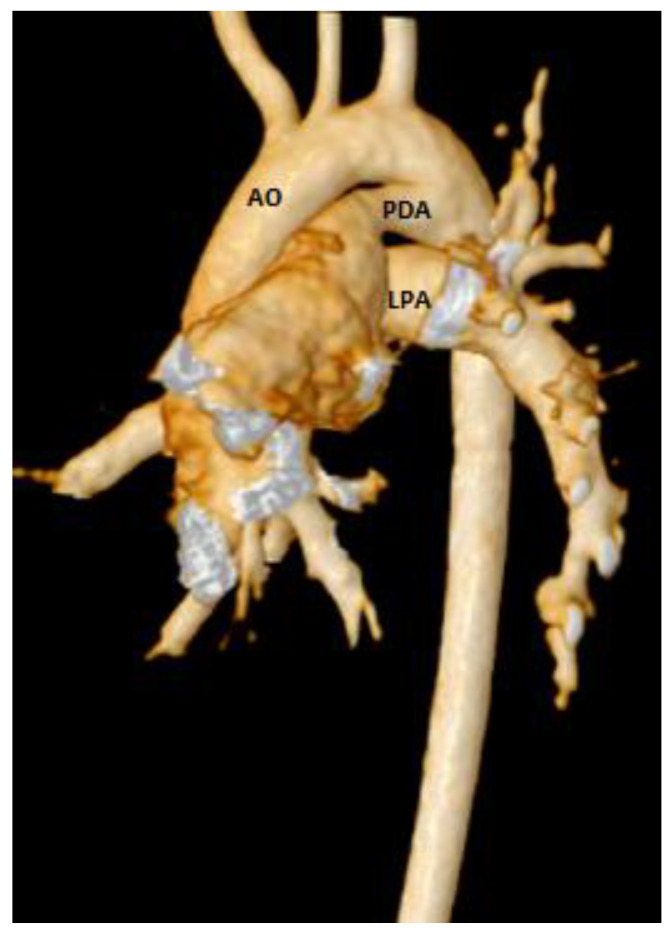
3D CMR reconstruction showing the aortic arch (AO), patent ductus arteriosus (PDA), and left branch pulmonary artery (LPA).

**Table 1 medicina-56-00636-t001:** Hemodynamic data obtained in our patient by right heart catheterization.

	Baseline (Room Air)	100% O_2_ + 40 ppm NO
RAP mm Hg (mean)	10/11 (9)	14/12 (11)
PAWP mm Hg (mean)	10/11 (10)	14/14 (13)
PAP mm Hg (mean)	79/43 (57)	90/46 (66)
AoP mm Hg (mean)	78/43 (59)	97/46 (65)
PA saturation %	84	97
CI L/min/m^2^	2.7	2.8
Qp L/min/m^2^	4.5	5.6
PVRi Wood U m^2^	10.4	9.4

(RAP = right atrial pressure, PAWP = pulmonary artery wedge pressure, PAP = pulmonary artery pressure, AoP = aortic pressure, PA = pulmonary artery, CI = cardiac index, PVRi = pulmonary vascular resistance index, NO = nitric oxide).

**Table 2 medicina-56-00636-t002:** Qp (pulmonary blood flow), PDA Flow, VSD Flow, pulmonary vascular resistance index (PVRi) as determined by iCMR.

	Baseline (Room Air)	100% O_2_ + 40 ppm NO
VSD Flow L/min/m^2^	1.8	2.5
PDA Flow L/min/m^2^	1.1	1.6
Qp L/min/m^2^	5.4	7.8
PVRi Woods U m^2^	8.7	6.8

PDA = patent ductus arteriosus, VSD = ventricular septal defect, iCMR = interventional cardiovascular magnetic resonance imaging.

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
