# Peer review of "Interventional Cardiovascular Magnetic Resonance Imaging (iCMR) in an Adolescent with Pulmonary Hypertension"

_medicina, 2020, doi:10.3390/medicina56120636_

Round 1

Reviewer 1 Report

This care report describes an adolescent male with PAH-CHD that after investigation of interventional CMR was changed from having an inoperable condition to an operable condition and underwent successful cardiac reconstructive surgery.

I have to main concerns with this case report.

The first major concern is that the interesting finding is how an investigation with iCMR contributed to change the status from inoperable to operable and this is not described in the paper. The reason for not operating should be the main focus in the introduction, the measures that changed this should be the main focus in the results (case report) and how the iCMR contributed to this change of mind should be the main focus of the discussion.

The second major concern is the need of a language check to improve readability and correctness of the message of the case report.

In addition, the number of references (31) is way too high for any case report. Please select and limit references to those of importance and relevance (i.e. modern) for this paper.

Introduction

Please rewrite and shorten the introduction substantially with the following in mind:

  • Echocardiography and right heart catheterization are not interchangeable. RV function cannot be assessed with RHC, and pressures cannot be accurately measured by Echo.
  • Complications of RHC is uncommon and is not the main reason for using cardiac MRI.
  • RHC cannot be used to assess RV chamber size and function or measure blood flow.

The last paragraph of the introduction is correct in facts, but need improved language.

Why was the patient considered inoperable? As this was changed, the facts on why are important.

Case report

Why was vasodilatation test performed? There is no indication for vasodilation test in patients with PAH-CHD.

Details on who did what is not of interest for the reader, please describe the procedure of the investigation in a level of detail that is necessary for the reader to understand the procedure but not more than that. I.e. the size of the sheet is of less interest, while the size of the catheter is…

The rest of the case report is confusing and a mix of methods and results to a degree that makes it almost non-readable. Please rewrite and have methods of RHC and CMR separate from the results.

Discussion

Main:

The discussion should focus on 1) what iCMR adds to performing RHC, CMR and Echo as separate investigations. Less radiation exposure is of course better for the patient, but there needs to be more than that to invest in this equipment and 2) what did iCMR add so the patient was now considered operable, compared to performing RHC, CMR and Echo as separate investigations.

Minor:

Is iCMR and CMR-RHC the same? Or do they differ?

The second paragraph is not relevant for this case report.

Positive vasodilatation test is a decrease in MPAP with unchanged or increased CO. PVR is not a main component of the determination of a positive or negative vasodilatation test.  In addition, vasodilation test is not indicated in patients with PAH-CHD.

The reference to lung biopsy being performed in patients with PVR > 8 WU is from 1989 and is not valid.

Figure 3 do not add anything to the case report. Please add a Figure showing intracardiac pressures measured with this equipment.

Author Response

Response to Reviewer-1:

This care report describes an adolescent male with PAH-CHD that after investigation of interventional CMR was changed from having an inoperable condition to an operable condition and underwent successful cardiac reconstructive surgery.

 I have two main concerns with this case report.

The first major concern is that the interesting finding is how an investigation with iCMR contributed to change the status from inoperable to operable, and this is not described in the paper. The reason for not operating should be the main focus in the introduction, the measures that changed this should be the main focus in the results (case report) and how the iCMR contributed to this change of mind should be the main focus of the discussion.

The second major concern is the need for a language check to improve readability and correctness of the message of the case report.

In addition, the number of references (31) is way too high for any case report. Please select and limit references to those of importance and relevance (i.e. modern) for this paper.

  • Thank you for your input. We abridged the Introduction section and significantly decreased the number of citations from 31 to 16.

Introduction

Please rewrite and shorten the introduction substantially with the following in mind:

  • Echocardiography and right heart catheterization are not interchangeable. RV function cannot be assessed with RHC, and pressures cannot be accurately measured by Echo.
  • Complications of RHC is uncommon and is not the main reason for using cardiac MRI.
  • RHC cannot be used to assess RV chamber size and function or measure blood flow.

  • Thank you. We revised the introduction including pro-and cons of RHC vs iCMR.

 The last paragraph of the introduction is correct in facts but need improved language.

  • Thank you. We revised the last paragraph.

Why the patient was considered inoperable? As this was changed, the facts on why are important.

  • The patient was deemed inoperable based on echocardiographic findings of bi-directional shunt by the referring physician. RHC determined conventional Fick principle PVRi >8 WU. m2 both at rest and with vasodilator challenge, but iCMR-determineed PVRi ≤ 8 WU. m2 and decided to proceed with VSD and PDA closure. This part is included in the revised manuscript.

 Case report

Why was vasodilatation test performed? There is no indication for vasodilation test in patients with PAH-CHD.

  • We disagree with the reviewer. The vasodilator testing in children with PH with CHD is strongly recommended based on current guidelines (last updated in 2018). (Ref: https://phassociation.org/wp-content/uploads/2018/02/Acute-Vasodilator-Testing_2018Final.pdf Accessed on 11/9/2020)
  • A second reference: Abman SH, Hansmann G, Archer S, et al. American Heart Association and American Thoracic Society Joint Guidelines for Pediatric Pulmonary Hypertension. Circulation. 132:2037-99, 2015.
  • Galiè N, Humbert M, Vachiery JL, et al. 2015 ESC/ERS Guidelines for the diagnosis and treatment of pulmonary hypertension. Eur Respir J 2015; 46: 903–975
  • These references are not included in the revised manuscript, as vasodilator testing in PH with CHD is standard.

Details on who did what is not of interest for the reader, please describe the procedure of the investigation in a level of detail that is necessary for the reader to understand the procedure but not more than that. I.e. the size of the sheet is of less interest, while the size of the catheter is…

  • Thank you. We revised the CMR based on RHC procedure.

The rest of the case report is confusing and a mix of methods and results to a degree that makes it almost non-readable. Please rewrite and have methods of RHC and CMR separate from the results.

  • Thank you. The case report is revised.

Discussion

Main:

The discussion should focus on 1) what iCMR adds to performing RHC, CMR and Echo as separate investigations. Less radiation exposure is of course, better for the patient, but there needs to be more than that to invest in this equipment and 2) what did iCMR add so the patient was now considered operable, compared to performing RHC, CMR and Echo as separate investigations.

  • Thank you. We clarified the role of CMR guided RHC and Echocardiography.

 Minor:

Is iCMR and CMR-RHC the same? Or do they differ?

  • iCMR has broad applications and encompasses CMR guided RCH. iCMR can estimate pulmonary and systemic blood flow and PVR accurately in the setting of multiple shunt lesions.

Positive vasodilatation test is a decrease in MPAP with unchanged or increased CO. PVR is not a main component of the determination of a positive or negative vasodilatation test.  In addition, vasodilation test is not indicated in patients with PAH-CHD.

  • We agree with the reviewer of the cardiac out component and disagree with the reviewer regarding the need for a vasodilator challenge test. Please refer to the explanation as above.

The reference to lung biopsy being performed in patients with PVR > 8 WU is from 1989 and is not valid.

  • Thank you, we removed this sentence and revised the discussion.

 Figure 3 do not add anything to the case report. Please add a Figure showing intracardiac pressures measured with this equipment.

  • See Reviewer-3 response. We left Figure-3 since it gives an idea of the laboratory set-up for all readers in general. Agree with the reviewer it may be redundant for experts in PH.

Authors describe the role of MRI -guided cardiac catheterization procedure to accurately estimate PVR in the setting of multiple shunt lesions (e.g. ventricular septal defects).

The proposal is a new and, in my point of view (as a pediatric cardiologist with interest in pediatric PH) highly interesting method.

  • Thank you.

Reviewer 2 Report

This presents a single patient with anatomy that would make quantification of the shunt hard to impossible to perform by catheterization. Utilization of MRI is undoubtedly an important method to quantify shunts. This additionally shows that MRI can be utilized to guide the catheterization. Though this is not a distinctly novel utilization of MRI, utilizing it in this manner has not been described. The combination of MR/cath was shown as a method to decrease radiation exposure while still obtaining the necessary information to guide care.

As a singular case report, this represents a novel presentation but requires a lab that most centers do not currently possess. However, additional reports like this continue to show the benefits of such an environment. It would be nice to know how much total anesthesia time was required for this procedure as a comparison to performing these procedures separately.

Author Response

Thank you very much.
I agree with the reviewer that this is novel, and hopefully, in future, this is adopted frequently in most centers in the world. iCMR software is available in all MRI scanners, but the protocol to use needs to be implemented.

As there is currently a learning curve regarding anesthesia, it is slightly more than a conventional cardiac catheterization. A previous study has reported that this balances out with experience (Ref #8) cited in the manuscript.

Reviewer 3 Report

Authors describe the role of MRI -guided cardiac catheterization procedure to accurately estimate PVR in the setting of multiple shunt lesions ( e.g. ventricular septal defects).

The propose a new and in my point of view (as a pediatric cardiologist with interest in pediatric PH) highly interesting method.

Major remarks:  None.

Minor remarks:

Introduction Section: --> Authors state ""Echocardiography is the most commonly used non-invasive tool for initial assessment and serial follow-up" but fail to provide a reference (e.g. J Heart Lung Transplant. 2019 Sep; 38(9): 879-901. doi: 10.1016/j.healun.2019.06.022// or: Pulm Circ. 2016 Mar; 6(1): 15-29. doi: 10.1086/685051).

Case Report: In general appropriate with extremly interesting figures!

Authors state ""Ventricular stroke volume (SV) was the difference between the EDV and ESV, and ventricular ejection fraction (%) was(SV/EDV) ×100"" and should provide an appropriate reference.

In my opinion the last part of the Case Report ""The VSD was closed with valved two-layer patch technique, primary closure of ASD and PDA was ligated and divided. His postoperative period was uncomplicated and discharged home with sildenafil and bosentan (targeted PH therapies)"" should be omitted, as the operation and follow up period is not important.

Discussion:

Authors find that ""Our experience in this case and many CHD patients [19]
demonstrate that iCMR can be safely performed in patients with adverse cardiopulmonary hemodynamics and incorporated into routine clinical practice without the need for expensive extra infrastructure"" --> as this is a very relevant information for the audience that paragraph can be pointed out in more detail.

Again, authors found that ""iCMR accurately determined the PVRi and demonstrated reversibility with acute vasodilator challenge, and the patient underwent surgical repair of VSD, and PDA"" --> this is a highly interesting finding and should be explained in  more detail for this reviewer and the audience of Medicina.

Author Response

Response to Reviewer-3: 

Authors describe the role of MRI -guided cardiac catheterization procedure to accurately estimate PVR in the setting of multiple shunt lesions ( e.g. ventricular septal defects).

The propose a new and in my point of view (as a pediatric cardiologist with interest in pediatric PH) highly interesting method.

Major remarks:  None.

Minor remarks:

Introduction Section: --> Authors state ""Echocardiography is the most commonly used non-invasive tool for initial assessment and serial follow-up" but fail to provide a reference (e.g. J Heart Lung Transplant. 2019 Sep; 38(9): 879-901. doi: 10.1016/j.healun.2019.06.022// or: Pulm Circ. 2016 Mar; 6(1): 15-29. DOI: 10.1086/685051).

  • Thank you. We added the reference: J Heart Lung Transplant. 2019 Sep; 38(9): 879-901 (Reference #3).

Case Report: In general appropriate with extremely interesting figures!

Authors state "Ventricular stroke volume (SV) was the difference between the EDV and ESV, and ventricular ejection fraction (%) was (SV/EDV) ×100"" and should provide an appropriate reference.

  • Thank you. We added the reference #9: Fratz S, Chung T, Greil G, et al. Guidelines and protocol for cardiovascular magnetic resonance in children and adults with congenital heart disease: SCMR expert consensus group on congenital heart disease. J Cardiovasc Magnet Resonance 2013;15:15-41

In my opinion, the last part of the Case Report ""The VSD was closed with valved two-layer patch technique, primary closure of ASD and PDA was ligated and divided. His postoperative period was uncomplicated and discharged home with sildenafil and bosentan (targeted PH therapies)"" should be omitted, as the operation and follow up period is not important.

  • Thank you. We removed this part from the case report and is not included in the revised manuscript.

Discussion:

Authors find that ""Our experience in this case and many CHD patients [19] demonstrate that iCMR can be safely performed in patients with adverse cardiopulmonary hemodynamics and incorporated into routine clinical practice without the need for expensive extra infrastructure"" --> as this is a very relevant information for the audience that paragraph can be pointed out in more detail.

  • Due to constraint for a case report, we have cited the original paper detailing our experience for readers interested in this topic (New Reference #4)

Again, authors found that "iCMR accurately determined the PVRi and demonstrated reversibility with acute vasodilator challenge, and the patient underwent surgical repair of VSD, and PDA"" --> this is a highly interesting finding and should be explained in more detail for this reviewer and the audience of Medicine.

  • Again, due to a case report constraint, we have cited our original paper and other relevant references (Reference #4-6) detailing iCMR experience for readers interested in this topic.

Reviewer 4 Report

The manuscript presents a case study combining MRI with RHC to evaluate pulmonary hemodynamics and morphology in a single 16 year old patient with congenital heart defects and pulmonary hypertension.

There are a few typos throughout the manuscript, but this was an overall well written case report. However, the Introduction did not adequately outline the need for this case report. Is this the first time such a procedure has been performed? Is there something specific about patients with these defects that require iCMR, or is the intention to limit radiation exposure? If the intention is to limit radiation exposure, that is an important goal, but a single patient case study seems insufficient. In that case, a broader study with statistical rigor would need to show accuracy, repeatability, and any other clinical advantages. The current case study obtained measurements, which the authors call “accurate” and “reproducible” in the discussion, but neither claim is supported. Maybe the fact that you showed PVRi responds to a vasodilator challenge does serve as some evidence of reliability, but it is not really an argument for accuracy or reproducibility.

The discussion also touches on the clinical logistics of this approach (costs, acquisition times), but additional justifications are needed that balance all of the aforementioned considerations between CMR+RHC against competing modalities.

Author Response

Response to Reviewer-4:

The manuscript presents a case study combining MRI with RHC to evaluate pulmonary hemodynamics and morphology in a single 16 year old patient with congenital heart defects and pulmonary hypertension.

There are a few typos throughout the manuscript, but this was an overall well written case report. However, the introduction did not adequately outline the need for this case report. Is this the first time such a procedure has been performed?

  • Thank you. We revised the introduction. We have cited our experience with iCMR in CHD (Ref #4). iCMR use in PH in children is not the first time, but only a few cases have been reported; it adds to the literature and justifies that accurate calculation of PVR is possible in the setting of multiple L>R shunts.

Is there something specific about patients with these defects that require iCMR, or is the intention to limit radiation exposure? If the intention is to limit radiation exposure, that is an important goal, but a single patient case study seems insufficient. In that case, a broader study with statistical rigor would need to show accuracy, repeatability, and any other clinical advantages. The current case study obtained measurements, which the authors call “accurate” and “reproducible” in the discussion, but neither claim is supported. Maybe the fact that you showed PVRi responds to a vasodilator challenge does serve as some evidence of reliability, but it is not really an argument for accuracy or reproducibility.

  • Agree with Reviewer. We have written that based on our case on multiple CHD cases (Ref #4), and in this case, we are presenting a case with PH where measurement of PVR and vasodilator testing is accurate and feasible.

The discussion also touches on the clinical logistics of this approach (costs, acquisition times), but additional justifications are needed that balance all of the aforementioned considerations between CMR+RHC against competing modalities.

  • Thank you. The advantages of iCMR are accurate estimation of pulmonary blood flow, calculation of PVRi, and no exposure to radiation. It is challenging to analyze the cost-effectiveness of iCMR vs. routine cardiac catheterization. However, there are already reports evaluating the total time to complete iCMR procedure vs. standard CMR (Ref#8). In future studies, it is possible to do a cost-effective analysis.